# Measurement and Calibration of DEM Parameters of Soybean Seed Particles

**Dongxu Yan [1], Jianqun Yu [2], Yang Wang [2], Kai Sun [2], Long Zhou [3], Ye Tian [4] and Na Zhang [2,\***

[1] Hua Lookeng Honors College, Changzhou University, Changzhou 213164, China
[2] School of Biological and Agricultural Engineering, Jilin University, Changchun 130022, China
[3] School of Agricultural Engineering and Food Science, Shandong University of Technology, Zibo 255022, China
[4] Center of Industry and Technology, Hebei University of Technology Petroleum, Chengde 067000, China
[\*] Correspondence: zna18@mails.jlu.edu.cn; Tel.: +86-138-449-41571

**Abstract:** In discrete element method (DEM) simulations, accurate simulation parameters are very important. For ellipsoidal soybean seed particles, the rolling friction coefficient between seed particles (RFCP-P) and the rolling friction coefficients between seed particle and boundary (RFCP-B) are difficult to measure experimentally and therefore need to be calibrated. In this paper, soybean seed particles of three varieties with different sphericities were taken as the research objects. Through the simulation analysis of repose angle and self-flow screening, it was shown that the above two parameters needed to be accurately calibrated. In addition, the sensitivity of the RFCP-P and RFCP-B to the angle of repose was analyzed by simulating the repose angle test. The results showed that the RFCP-P had a significant effect on the test results of the repose angle, and the RFCP-B had little effect on the test results of the repose angle. Therefore, the RFCP-P was calibrated using a single-factor test of repose angle, and the RFCP-B was calibrated using the repose angle test with soybean particles mixed with organic glass spheres. The accuracy of the calibration parameters was verified by rotating cylinder test and self-flow screening test.

**Keywords:** discrete element method; soybean seed; ellipsoidal shape; parameter calibration; rolling friction coefficient

## 1. Introduction

The DEM is widely used in the field of agricultural engineering, and its parameters are crucial to the simulation results [1–10]. For soybean seed particles, parameters play a crucial role in the simulation results when DEM is used to analyze the movement of the particles, the contact interaction between the particles, and between the particles and the boundary.

Nguyen, et al. [11] studied soybean seed particles of one variety, which were approximated as being spherical. The physical properties of the soybean (particle size distribution and weight properties) and the static friction coefficient between the particles and the material surface were determined by test. The rest of the DEM simulation parameters were calibrated by different particle flow tests. In fact, with such a calibration method, there are multiple combinations of parameters that meet the requirements. The applicability of such parameter results needs further analysis when the calibration is performed without determining whether the calibration test is sensitive to only one of the parameters.

Bhupendra et al. [12] used a spherical soybean seed particle as their study object. A set of calibration results were obtained by stacking tests, as follows: the restitution coefficient, static friction coefficient, and rolling friction coefficient between the particles, and the restitution coefficient, static friction coefficient, and rolling friction coefficient between the particles and the boundary. However, the restitution coefficient and static friction coefficient between the particles obtained by calibration were quite different from the actual values. In order to make the simulation closer to the test and accurately analyze the particle

population movement, the restitution coefficient and static friction coefficient between the particles can be determined by the test method.

Some scholars [13,14] believe that non-spherical particles do not need to consider the rolling friction coefficient, and only need to calibrate the corresponding sliding friction coefficient to meet testing requirements. However, others believe that the effect of rolling friction coefficient on the test results is significant even for non-spherical particles [15–17]. For soybean seed particles of ellipsoidal shape [18,19], how much the rolling friction coefficient affects the particle population motion and whether the rolling friction coefficient needs to be accurately calibrated requires further study.

Long Zhou [15] demonstrated that the rolling friction coefficient has a significant effect on test results using sensitivity tests during the modeling of different shapes of corn seeds, and further calibrated the rolling friction coefficients between particles, and between particles and boundaries through a piling angle test. For soybean seed particles, the sensitivity of the rolling friction coefficient to the test results needs to be determined further, and the calibration method of the sensitivity parameter also needs to be studied in-depth.

Based on the above problems, in this paper, we verified the importance of the parameters for the first time, and a method for calibrating the above simulation parameters was proposed and verified by tests. Three representative soybean varieties, SN42, JD17, and ZD39, were used in this study. Some parameters of soybean seed particles were measured by test methods. The effect of RFCP-P and RFCP-B on powder motion was analyzed by a repose angle test and self-flow screening simulation. After analyzing the sensitivity of parameters, the test method of parameter calibration was determined and parameter values were calibrated. The accuracy of the parameters was verified using a rotating cylinder test and self-flow screening test. This paper provides some reference for the calibration of parameters for soybean seed particles.

## 2. Measurement of Soybean Seed Particle Parameters

In this section, the moisture content, triaxial dimensions, particle density, stiffness coefficient, elasticity modulus, restitution coefficient, and static friction coefficient of soybean seed particles are measured by test measurements; Poisson's ratio is taken to be 0.4, according to reference [20].

### 2.1. The Moisture Contents of Soybean Seed Particles

The moisture content of soybean seed particles was measured using XY-102MW type halogen moisture meter (accuracy 0.001), and the test was repeated 5 times for each variety, The moisture content was 10.31, 8.08, and 11.1% for SN42, JD 17, and ZD39, respectively.

### 2.2. Particle Density of Soybean Seed Particles

In this paper, the particle density of soybean seeds were measured by the pycnometer method, and the calculation formula is as follows:

$$\rho_0 = m_0 / V_0 \tag{1}$$

where $\rho_0$ is the density of soybean seed particles, g/cm$^3$; $m_0$ is the mass of soybean seeds particles, g; $V_0$ is the volume of soybean seed particles, cm$^3$. The formula for the volume of soybean particles is as follows:

$$V_0 = \frac{(m_2 - m_1)(m_3 - m_1 - m_0)}{\rho_w} \tag{2}$$

where $m_1$ is the mass of the dry specific gravity bottle, g; $m_2$ is the mass of the specific gravity bottle filled with water, g; $m_3$ is the mass of soybean seed particles, water, and specific gravity bottle, g; and $\rho_w$ is the density of water, g/cm$^3$.

### 2.3. Stiffness Coefficients of Soybean Seed Particles

The stiffness coefficients of the three varieties were measured using the compression test method [21,22]. As soybean seed particles are ellipsoidal particles with three unequal axes, their stiffness coefficients are different in all directions. Test measurements with three different placement methods (horizontal, lateral, and vertical) are shown in Figure 1. The average value was obtained and used as its final stiffness coefficient.

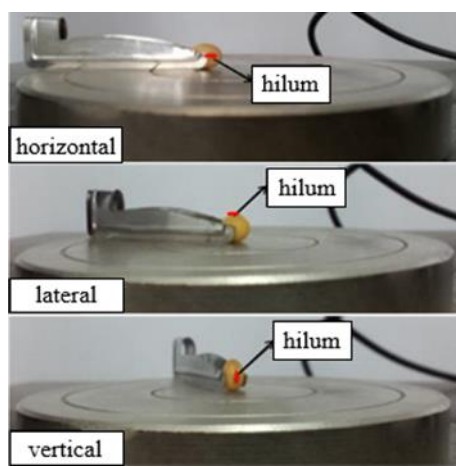

**Figure 1.** Placement of soybean seed particles.

Taking SN42 as an example, during the loading process, the test force was gradually increased with an increase in deformation before the abrupt change of the test force. The curve is divided into three sections for discussion. In the first section, the deformation (0–0.01 mm) is very small, the test force is small and the growth trend is not obvious; in the second section, the deformation (0.01–0.05 mm) is small, and the test force increases slowly in this range; in the third section, the deformation (0.05–0.4 mm)is larger, and the test force basically grows linearly, as shown in Figure 2.

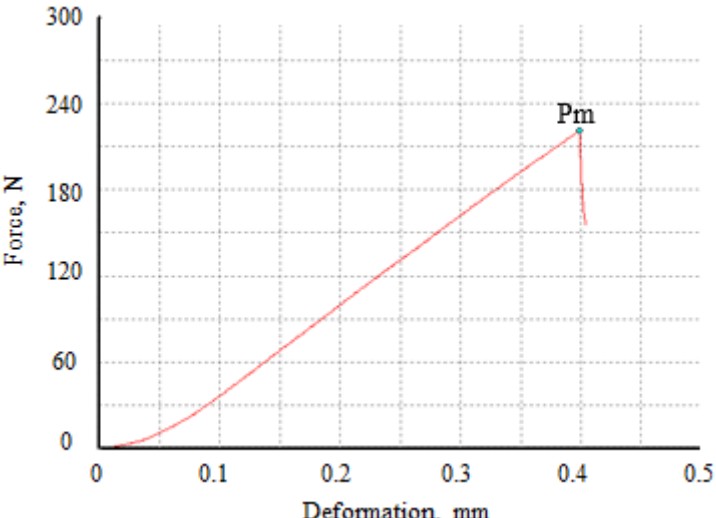

**Figure 2.** The force–deformation relationship of SN42 soybean seeds.

In our study, the deformation caused by collision during particle movement were within a small range of changes (0.01–0.05 mm) [21]. Therefore, the second section of the curve in the range of smaller deformations was analyzed and processed in this paper. This segment of data was processed in an Excel sheet and a straight line was fitted, as

shown in Figure 3. The slope of this straight line is the stiffness coefficient of the soybean seed particles.

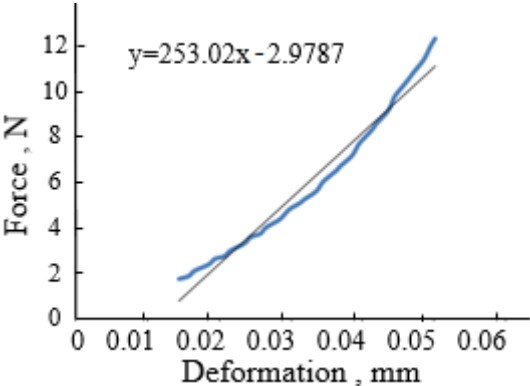

**Figure 3.** The force–deformation relationship of SN42 soybean seeds when the deformation is small.

*2.4. Elastic Modulus of Soybean Seed Particles*

The elasticity modulus of soybean seed particles was measured by compression tests [23]. Soybean seed particles are approximately ellipsoidal in shape, so the radii of curvature of soybean seed particles in contact with the upper and lower surfaces of the platen are the same. According to the standard ASAE S368.4 DEC2000 (R2008) [20], the elasticity modulus was calculated as follows:

$$E = \frac{0.338F(1-\mu^2)}{D^{3/2}} \left[ 2K_U \left( \frac{1}{R} + \frac{1}{R'} \right)^{1/3} \right]^{3/2} \tag{3}$$

where $E$ is the modulus of elasticity of soybean seed particles, Pa; $D$ is the amount of deformation, mm, which is the middle value of the deformation corresponding to the previous measurement of the stiffness coefficient; $F$ is the test force corresponding to the current deformation, N, which can be directly found in the Excel database; $\mu$ is the Poisson's ratio, with a value of 0.4; $R$ and $R'$ are the primary and secondary curvature radii when soybean seed particles come in contact with the surface of the plate, m; and $K_U$ is the coefficient.

*2.5. Restitution Coefficient of Soybean Seed Particles*

2.5.1. Restitution Coefficient between Soybean Seed Particles and Boundary

A drop test [24,25] was used to measure the restitution coefficient between the soybean seed particle and boundary. Using SN42 as an example, the soybean seed particle is placed vertically at the vacuum nozzle along its length and marked directly in front of the seed particle, as shown in Figure 4a–c. A high speed camera is used to record the above test procedure, while ensuring that the soybean seed particle is moving in a vertical direction. The software accompanying the high-speed camera was used to analyse the experimental results. Neglecting air resistance, the restitution coefficient was calculated as follows:

$$e = \sqrt{\frac{h_1}{h_0}} \tag{4}$$

where $h_0$ is the distance between the initial position and coordinate origin of soybean seed particles, as shown in Figure 4b, and $h_1$ is the distance between the highest point and coordinate origin of the first vertical rebound of soybean seed particles, as shown in Figure 4c.

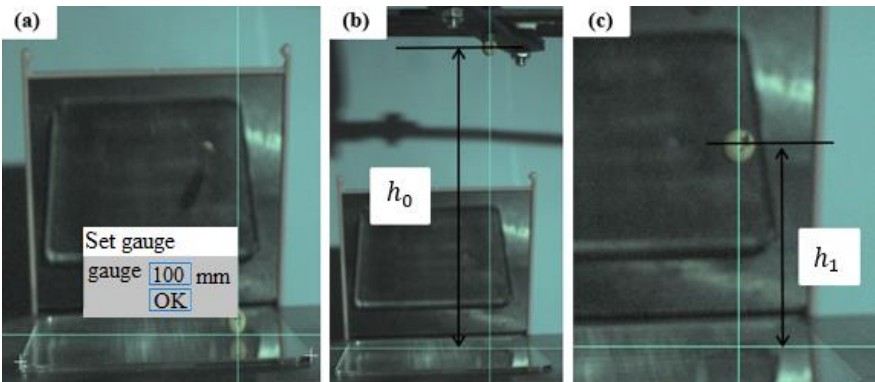

**Figure 4.** PCC image analysis process: (**a**) coordinate origin selection and distance calibration, (**b**) the distance between the initial position and coordinate origin of soybean seed particles, and (**c**) the distance between the highest point and coordinate origin of the first vertical rebound of soybean seed particles.

2.5.2. Restitution Coefficient between Soybean Seed Particles

A single pendulum collision test [26,27] was used to measure the restitution coefficient between soybean seed particles. A soybean was lifted with a straightedge to the position shown in Figure 5a–c. The software that accompanied the high-speed camera was used to analyse the experimental results. The equation for the restitution coefficient, ignoring air resistance, was as follows:

$$e = (\sqrt{h_1} - \sqrt{h_2})/\sqrt{h_0} \tag{5}$$

where $h_0$ is the vertical distance between the initial position of soybean seed particles and coordinate origin, as shown in Figure 5b, and $h_1$ and $h_2$ are the distances between the highest point of the vertical rebound of the two soya beans and the origin of the coordinate after the first collision, respectively, as shown in Figure 5c.

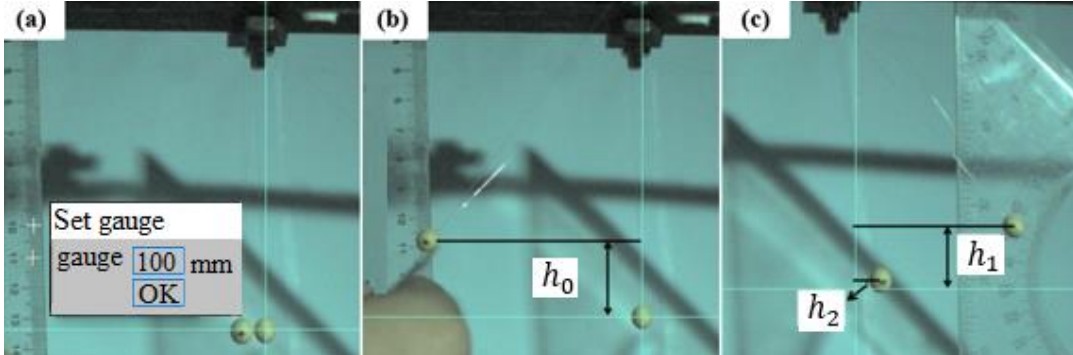

**Figure 5.** PCC image analysis process: (**a**) coordinate origin selection and distance calibration, (**b**) the vertical distance between the initial position of soybean seed particles and coordinate origin, (**c**) the distances between the highest point of the vertical rebound of the two soya beans and the origin of the coordinate after the first collision.

*2.6. Static Friction Coefficient of Soybean Seed Particles*

2.6.1. Static Friction Coefficient between Soybean Seed Particles and the Boundary

The method for measuring the static friction coefficient between the particles and boundary was the slope method [28,29]. Three soybean seed particles with an intact appearance were fixed on a small square glass plate with glue. The boundary material and inclinometer were fixed to the slope meter and the test material was placed on the boundary material, as shown in Figure 6. For the trials, three test specimens of each variety were used to produce three replicate experiments for each specimen. The specimens were

made with the soybean in a random orientation with no fixed direction. The formula for the coefficient of static friction between the soya seed pellet and the boundary is shown below:

$$\mu_{SP-B}= \tan\alpha \tag{6}$$

where $\mu_{SP-B}$ is the static friction coefficient between the soybean seed particle and boundary and $\alpha$ is the indication of the inclinometer when the test specimen is just sliding, rad.

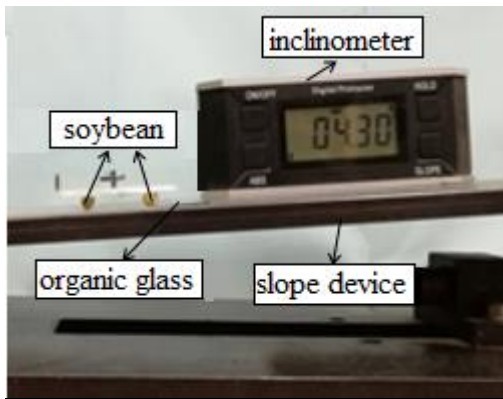

**Figure 6.** Static friction coefficient between soybean seed particle and boundary measured by the slope method.

2.6.2. Static Friction Coefficient between Soybean Seed Particles

The slope method was used to measure the static friction coefficient between soybean seed particles. The test specimens were made by fixing three intact appearing soybean seed particles to a small square glass piece with glue. We fixed one specimen on the inclinometer and placed the other on top, to ensure that the two test soybean particles were in exact vertex contact, as shown in Figure 7. The equation for calculating the static friction coefficient between soybean seed particles is as follows:

$$\mu_{SP-P}= \tan\beta \tag{7}$$

where $\mu_{SP-P}$ is the static friction coefficient between soybean seed particles and $\beta$ is the indication of the inclinometer when the test specimen is just sliding, rad.

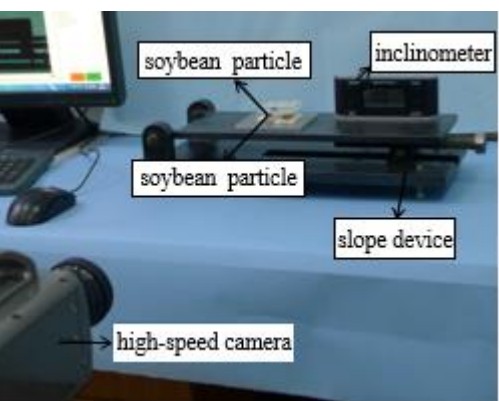

**Figure 7.** Static friction coefficient between soybean seed particles measured by the slope method.

2.7. *Soybean Seed Particle Model and Simulation Parameters*

It was clear from our previous work that geometric models of soybean seed particles with different sphericities can be built by 5-, 9-, and 13-sphere models [19]. In this paper,

two hundred grains of each variety were selected and measured to obtain the average triaxial dimensions, as well as the sphericity of soybean seed particles, as shown in Table 1.

**Table 1.** Triaxial dimensions of soybean seed particles of three varieties.

| Variety | Length, mm | Width, mm | Thickness, mm | Sphericity, % |
|---------|------------|-----------|---------------|---------------|
| SN42 | 7.44 | 7.24 | 6.51 | 94.78 |
| JD17 | 6.95 | 6.2 | 5.11 | 86.86 |
| ZD39 | 7.36 | 6 | 4.73 | 80.6 |

The 13-sphere model was chosen for the simulation, as shown in Figure 8, with powder generation performed according to volume normal distribution; the simulation parameters are shown in Table 2.

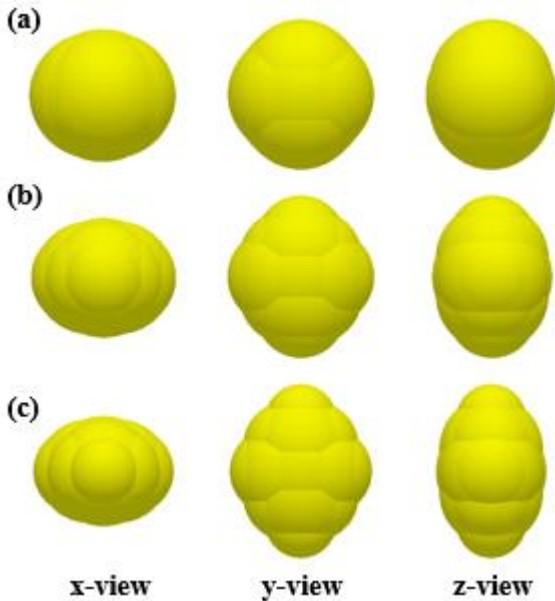

**Figure 8.** The 13-sphere model of (**a**) SN42, (**b**) JD17, and (**c**) ZD39.

**Table 2.** Parameters used in the simulation.

| Parameters | Symbol | SN 42 Soybean Seeds | SN 42 Organic Glass | JD 17 Soybean Seeds | JD 17 Organic Glass | ZD 39 Soybean Seeds | ZD 39 Organic Glass |
|------------|--------|---------|---------|---------|---------|---------|---------|
| Density, kg/m$^3$ | $\rho$ | 1257 | 1800 | 1213 | 1800 | 1192 | 1800 |
| Poisson's ratio | $\nu$ | 0.4 | 0.25 | 0.4 | 0.25 | 0.4 | 0.25 |
| Elastic modulus, Pa | $E$ | $7.59 \times 10^8$ | $1.30 \times 10^8$ | $6.07 \times 10^8$ | $1.30 \times 10^8$ | $2.55 \times 10^8$ | $1.30 \times 10^8$ |
| Static friction coefficient | $\mu_S$ | 0.205 | 0.228 | 0.211 | 0.228 | 0.207 | 0.235 |
| Restitution coefficient | $e$ | 0.627 | 0.542 | 0.562 | 0.642 | 0.607 | 0.705 |

## 3. Analysis of the Influence of the Rolling Friction Coefficient

This section analyses the influence of the RFCP-P on the angle of repose by means of a repose angle test and the influence of the RFCP-B on the percentage passing by means of a self-flow screening test. The results confirmed that accurate calibration of the above two parameters is necessary.

### 3.1. Repose Angle Simulation

The RFCP-B was set to 0.025 and the RFCP-P was set to nine values of 0, 0.025, 0.05, 0.075, 0.1, 0.125, 0.15, 0.175, and 0.2. The single factor test was conducted to analyze the effect of the RFCP-P on the angle of repose.

The test procedures were same as in previous studies; the dimensions of the device were 220*48 mm [19]. The EDEM (Version, 2018, School of Biological and Agricultural Engineering, Jilin University, Changchun, Jilin, China) software with the parameters in Table 1 was used to simulate the repose angle test. First, 4000 particles were generated in the particle factory. After two seconds, the insert plate was pulled out and the seed particles flowed out of the loading box; at the same time, the angle of repose was formed in the loading box. The repose angle results were analyzed using image processing software, as shown in Figure 9. The simulation tests were repeated five times for SN42, JD17, and ZD39, respectively, according to the above steps.

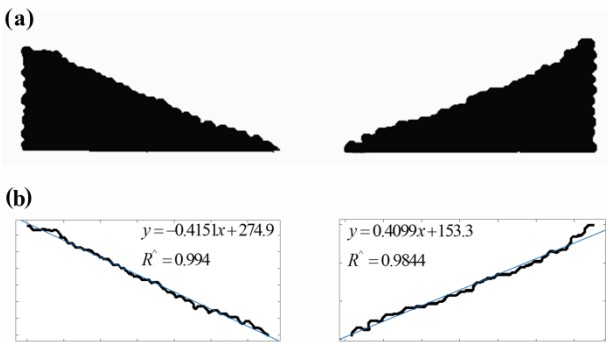

**Figure 9.** Image of (**a**) repose angle simulation results and (**b**) processing diagram.

### 3.2. Self-Flow Screening Simulation

The RFCP-P was set to 0.025 and the RFCP-B was set to nine level values of 0, 0.025, 0.05, 0.075, 0.1, 0.125, 0.15, 0.175, and 0.2, respectively. A single factor test was conducted to analyze the effect of the RFCP-B on the percentage passing.

The test procedures were the same as in previous studies [19].The inclination angle and aperture sizes of the three varieties of self-flow screening simulation were 11° and 8 mm, respectively. The EDEM software with the parameters in Table 1 was used to simulate the self-flow screening test. In the simulation, 1000 particles were generated in the particle factory. After the particles were stable, we pulled out the insert plate, and the soybean particles moved downward along the sieve. At the end of the movement, the particles were present in the receiving area below or on the sieve deck. The numbers of soybean particles in the corresponding areas were counted and numbered as 1–5, as shown in Figure 10.

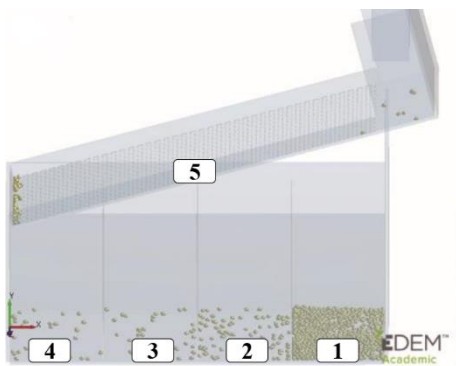

**Figure 10.** The statistical areas of self-flow screening simulation.

### 3.3. Analysis of Simulation Results

The relationship between the angle of repose and RFCP-P is shown in Figure 11. With the increase in RFCP-P from 0 to 0.2, the increasing trend of the angle of repose of the three varieties was obvious. For SN42, the angle of repose gradually increased from 19.33 to 37.57°; for JD17, the angle of repose gradually increased from 24.8 to 37.02°; and for ZD39, the angle of repose gradually increased from 27.31 to 39.1°. Analysis of the results shows that the RFCP-P had a significant effect on the angle of repose. Therefore, an accurate RFCP-P needs to be calibrated through calibration tests.

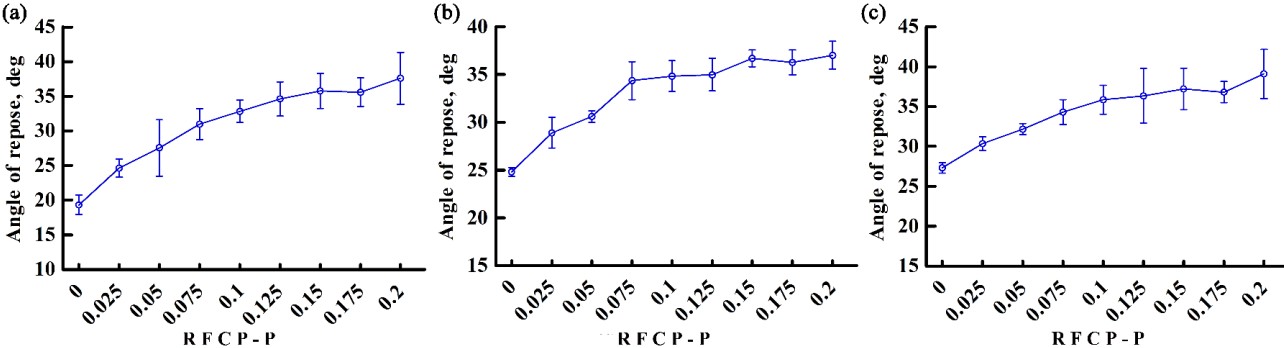

**Figure 11.** The relationship between angle of repose and RFCP-P for (**a**) SN42, (**b**) JD17, and (**c**) ZD39.

The relationship between percentage passing and RFCP-B is shown in Figure 12. For SN42, due to its high sphericity, the percentage passing did not change much when the RFCP-B gradually increased from 0 to 0.075, whereas the percentage passing tended to significantly decrease when the RFCP-B gradually increased from 0.075 to 0.2, with a range of 94.53 to 29.23%. For JD17 and ZD39, the percentage passing was significantly reduced as the RFCP-B increased, with a range of 95.43 to 26.4% and 86.97 to 23.53%, respectively. The analysis showed that the RFCP-B had a significant effect on the percentage passing. Therefore, an accurate RFCP-B needs to be calibrated by calibration tests.

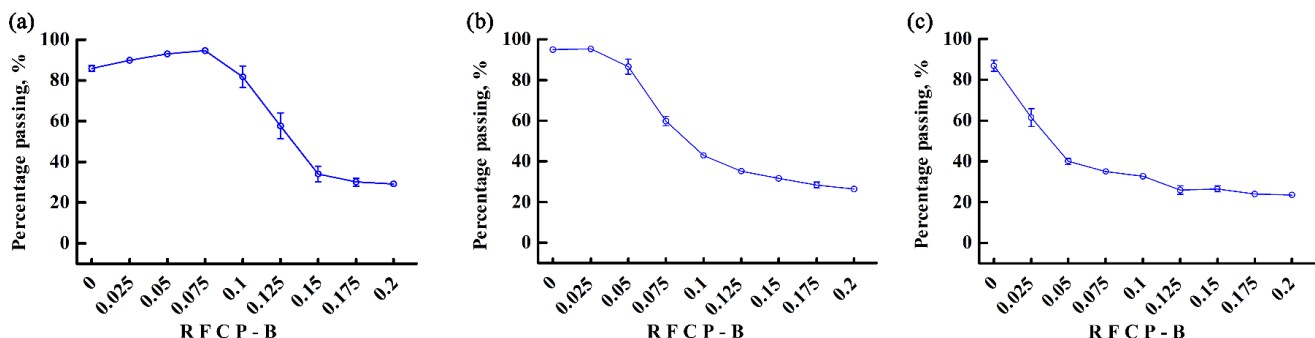

**Figure 12.** The relationship between percentage passing and RFCP-B for (**a**) SN42, (**b**) JD17, and (**c**) ZD39.

## 4. Study on the Sensitivity of RFCP-P and RFCP-B

### 4.1. Comprehensive Simulation Test of Sensitivity Analysis

As mentioned before, the RFCP-P and RFCP-B need to be calibrated. If two parameters are simultaneously calibrated, there are multiple sets of solutions. Therefore, the sensitivity of RFCP-P and RFCP-B to the angle of repose was first analyzed by simulating pitch angle tests. Seven levels of 0.01–0.07 were taken for both RFCP-P and RFCP-B, and a full 7*7 simulation was performed.

### 4.2. Analysis of the Results

The results of the effect of RFCP-P and RFCP-B on the angle of repose for the three varieties of soybean seed particles are shown in Figure 13. For the three varieties, the effect of RFCP-P on the angle of repose was highly significant, whereas the effect of RFCP-B on the angle of repose was insignificant.

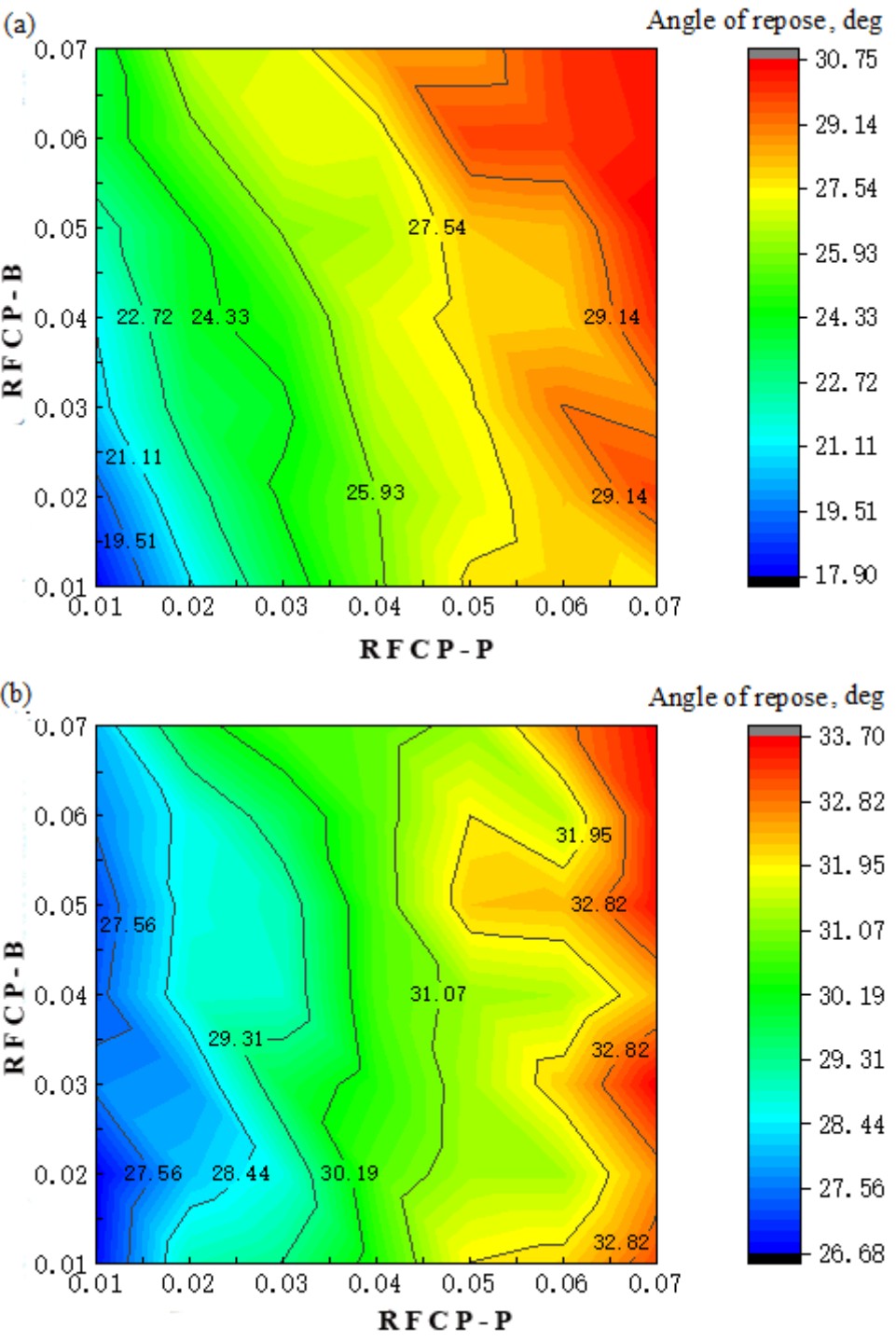

**Figure 13.** *Cont.*

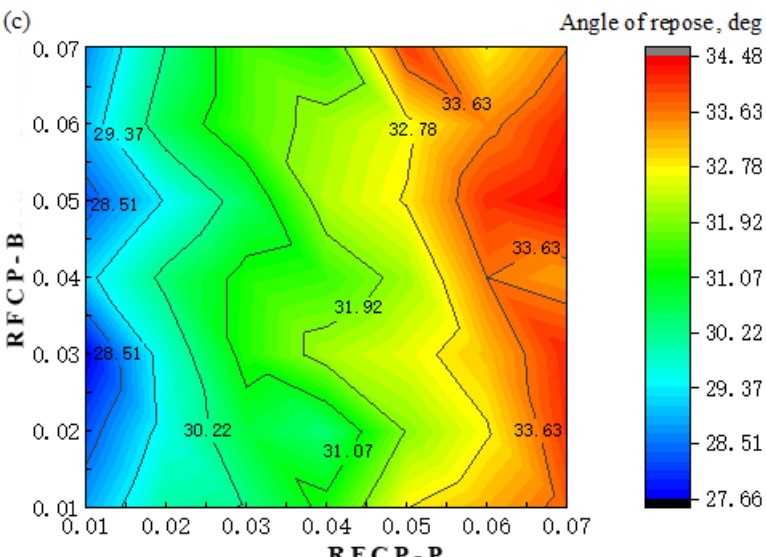

**Figure 13.** Sensitivity analysis of RFCP-P and RFCP-B to angle of repose for (**a**) SN42, (**b**) JD17, and (**c**) ZD39.

According to the above analysis, the RFCP-P can be calibrated by a single factor test of the repose angle.

## 5. Calibration of the Rolling Friction Coefficient

### 5.1. Calibration of the RFCP-P

From the previous analysis, the angle of repose was only sensitive to the RFCP-P, so the RFCP-P was calibrated by simulation of the repose angle. The RFCP-P was taken as 0.01, 0.02, 0.03, 0.04, and 0.05, and the RFCP-B was taken as 0.02 for the single factor test of the repose angle. The RFCP-P was calibrated by comparing the simulation and test results.

### 5.2. Calibration of the RFCP-B

After the calibration of the RFCP-P, the repose angle test was also used to calibrate the RFCP-B. In this paper, spheres with the same boundary material were processed, with a radius of 5 mm. The soybean seed particles and organic glass balls were 350 g each. They were uniformly mixed and poured into the loading box for testing, and the repose angle was simulated and analyzed, as shown in Figure 14. The rolling friction coefficient between the soybean seed particles and organic glass spheres obtained from the calibration was the RFCP-B.

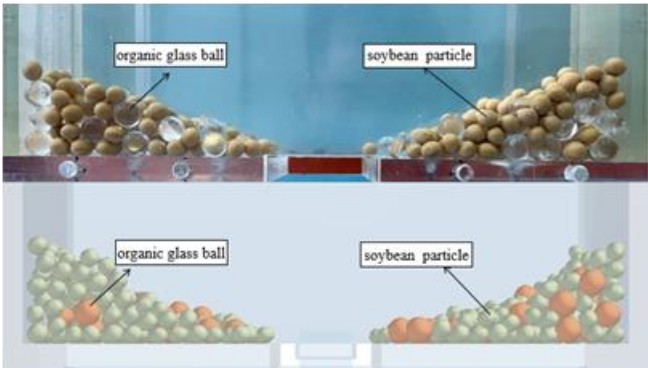

**Figure 14.** Test and simulation diagram of the repose angle of soybean seed particles mixed with organic glass spheres.

The static friction coefficient and rolling friction coefficient between the organic glass spheres and boundary involved in the simulation were measured by the slope method, and the restitution coefficient was measured by the drop test.

### 5.3. Analysis of Results

5.3.1. Calibration Results of the RFCP-P

The relationship between the angle of repose and the RFCP-P for the three varieties are shown in Figure 15.

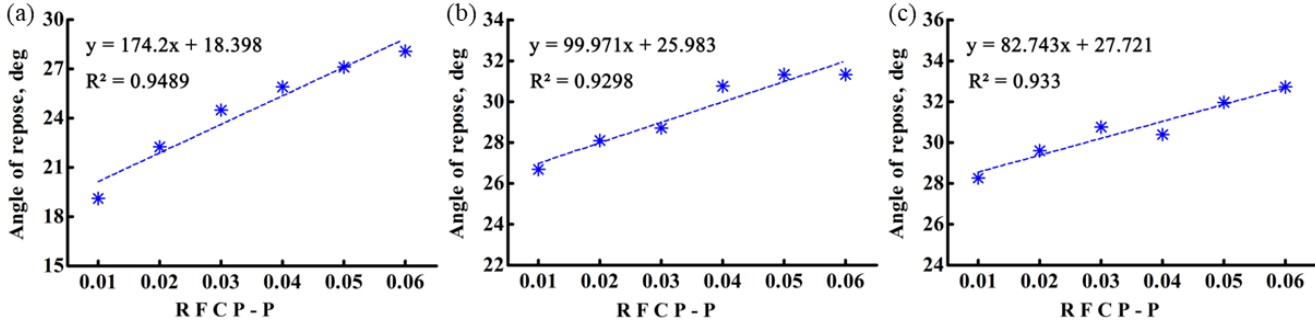

**Figure 15.** The relationship between the angle of repose and the RFCP-P for (**a**) SN42, (**b**) JD17, and (**c**) ZD39.

For the three varieties, the angle of repose tended to increase as the RFCP-P increased. The repose angle results for the three varieties (SN42, 23.86°; JD17, 27.78°; and ZD39, 38.97°.) were obtained by measuring the repose angle test of the soybean seed particles. Taking SN42 as an example, the relationship between the angle of repose and RFCP-P was obtained by liner fitting, and the formula is shown as follows.

$$y = 174.2x + 18.398 \ (R^2 = 0.9298) \tag{8}$$

The result (23.86°) of the repose angle test for SN42 was entered into Formula 8, and the RFCP-P was calculated to be 0.031. The same method was used to calculate the RFCP-P, which was 0.018 and 0.136 for JD17 and ZD39, respectively.

5.3.2. Calibration Results for RFCP-B

The relationship between the angle of repose formed by mixing the soybean seed particles with the organic glass spheres and rolling friction coefficient between the soybean seed particles and boundary (organic glass spheres) is shown in Figure 16.

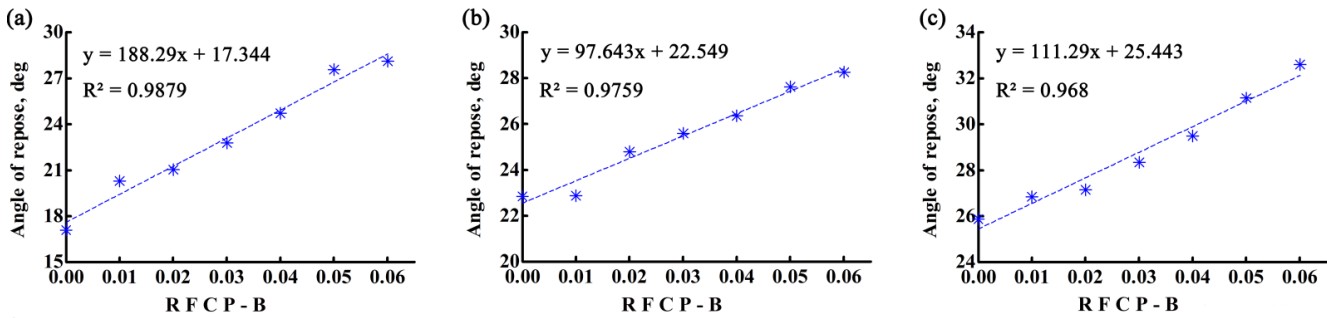

**Figure 16.** The relationship between the angle of repose and the RFCP-B for (**a**) SN42, (**b**) JD17, and (**c**) ZD39.

For the three varieties, the angle of repose tended to increase as the RFCP-B increased. The repose angle results for the three varieties (SN42, 18.72°; JD17, 26.16°; and ZD39, 28.56°.) were obtained by measuring the repose angle test of the soybean–glass ball mixture.

Taking SN42 as an example, the relationship between the angle of repose and RFCP-B was obtained by liner fitting, and the formula is shown as follows.

$$y = 188.29x + 17.344 \ (R^2 = 0.9879) \tag{9}$$

The result (18.72°) of the repose angle test for SN42 was entered into the Formula 9, and the RFCP-B was calculated to be 0.008. The same method was used to calculate the RFCP-B, which was 0.037 and 0.028 for JD17 and ZD39, respectively.

## 6. Test Verification of Calibration Parameters

The accuracy of the calibration parameters was verified by rotating cylinder test and self-flow screening test. At the same time, the difference between the test results and simulation results when the rolling friction coefficient was ignored was also analyzed.

### 6.1. Rotating Cylinder Test

The rotating cylinder test apparatus is shown in Figure 17a, where the inner diameter of the cylinder is 160 mm and the height is 40 mm. Taking SN42 as an example, the soybean seed particles were first filled through the inlet, and when the height of the soybean seed particle filling in the cylinder reached 40–50 mm, it met the particle filling requirements of the rotating cylinder test, as shown in Figure 17b.

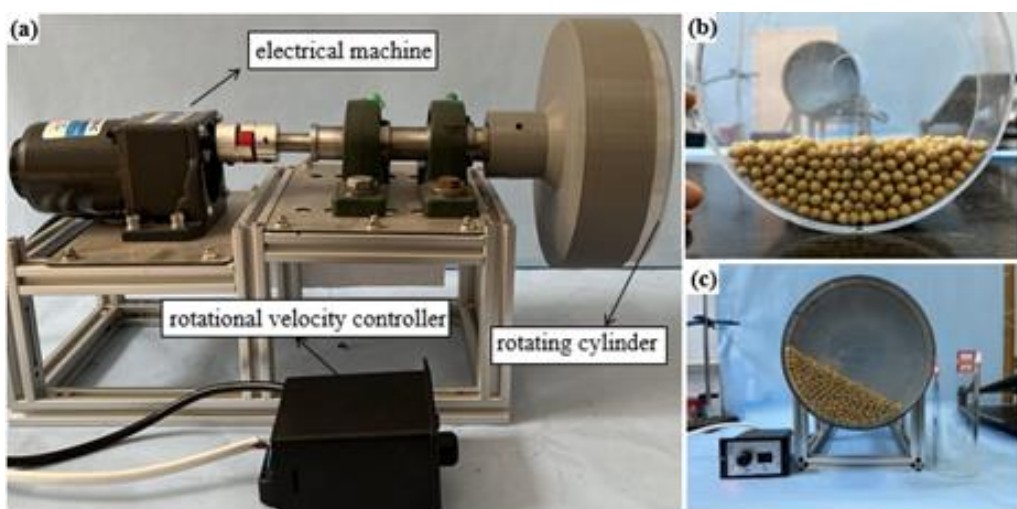

**Figure 17.** (**a**) The rotating cylinder test apparatus and (**b**,**c**) the screenshot of test process.

After the test device was installed, the power supply was connected, and the speed controller was adjusted to make the drum speed 7.5 rpm. After the speed was stable, the soybean seed particles in the drum formed a dynamic repose angle, such as Figure 17c, recording the dynamic piling process for 30 s. The cylinder speed was adjusted to 11.5 and 15.5 rpm, respectively, and the dynamic stacking process was recorded for 30 s. At the end of the test, the soybean seed particles were weighed for mass. Three replicate tests were conducted for each variety.

For the simulated rotating cylinder test, the soybean seed particle model was equal to the actual soybean seed particle mass. Taking SN42 as an example, the drum speed was 7.5 rpm. Figure 18a,b shows the test photo and binarization image, respectively. Figure 18c,d shows the simulation screenshot using the calibration parameters and binarization image, respectively. Figure 18e,f shows the simulation screenshot without considering the rolling friction coefficient and binarization image, respectively.

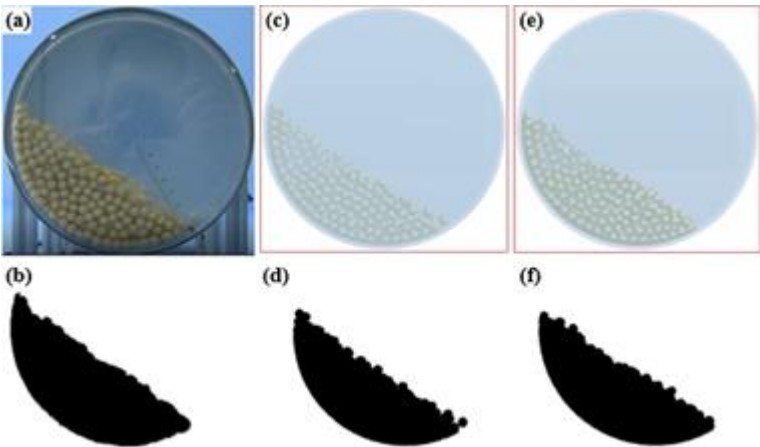

**Figure 18.** (**a**) The test photo of the rotating cylinder and (**b**) its binarization image; (**c**) simulation result snapshot using calibration parameters and (**d**) its binarization image; and (**e**) simulation result snapshot without considering rolling friction coefficient and (**f**) its binarization image.

Figure 19 shows the dynamic angle of repose versus cylinder speed for each variety. For SN42, the relative errors between the simulation results using the calibration parameters and experimental results were 2.28, 6.93, and 4.78% for rotational speeds of 7.5, 11.5, and 15.5 rpm, respectively, whereas the relative errors between the simulation results ignoring the rolling friction coefficient and experimental results were 13.85, 17.08, and 12.83% for rotational speeds of 7.5, 11.5, and 15.5 rpm, respectively, as shown in Figure 19a.

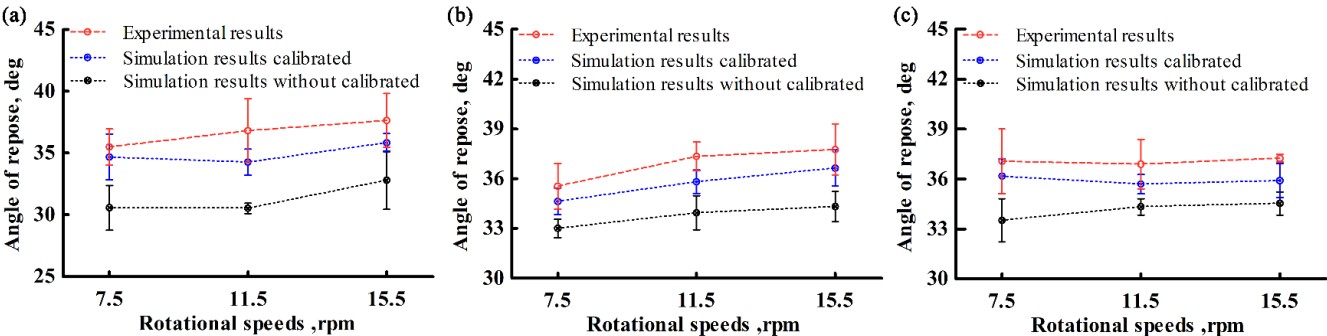

**Figure 19.** The relationship between dynamic angle of repose and rotational speed for (**a**) SN42, (**b**) JD17, and (**c**) ZD39.

For JD17, the relative errors between the simulation results using the calibration parameters and experimental results were 2.56, 4.1, and 2.84% for rotational speeds of 7.5, 11.5, and 15.5 rpm, respectively, whereas the relative errors between the simulation results ignoring the rolling friction coefficient and experimental results were 7.15, 9.11, and 9.11% for rotational speeds of 7.5, 11.5, and 15.5 rpm respectively, as shown in Figure 19b.

For ZD39, the relative errors between the simulation results using the calibration parameters and experimental results were 2.43, 3.25, and 3.62% for rotational speeds of 7.5, 11.5, and 15.5 rpm, respectively, whereas the relative errors between the simulation results ignoring the rolling friction coefficient and experimental results were 9.55, 6.94, and 7.28% for rotational speeds of 7.5, 11.5, and 15.5 rpm, respectively, as shown in Figure 19c.

The analysis shows that the relative error between the simulation results using calibration parameters and experimental results was small and within the error range of the experimental results. The simulation results without taking the rolling friction coefficient into account were much smaller than the experimental results, and were not within the experimental error range.

### 6.2. Self-Flow Screening Tests

According to previous studies, the inclination angles of the self-flow screening devices were 7, 11, and 15° for SN42, JD17, and ZD39, respectively. The material of the device was organic glass and the aperture size was 8 mm. Figure 20a–c shows the test photo for self-flow screening, the simulation screenshot using the calibrated parameters, and the simulation screenshot without considering the rolling friction coefficient, respectively.

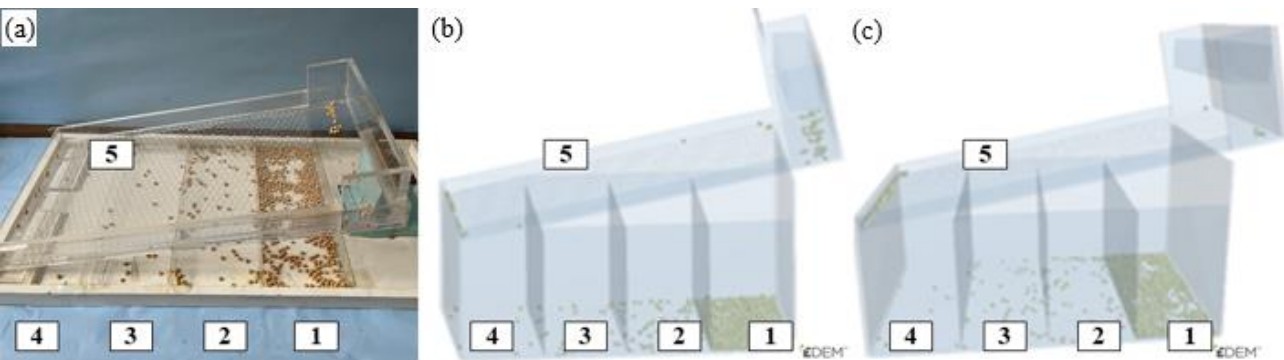

**Figure 20.** (**a**) The test photo, (**b**) the simulation screenshot using the calibrated parameters, and (**c**) the simulation screenshot without considering the rolling friction coefficient for self-flow screening test.

Figure 21a–c shows comparisons of the simulation results with experimental results of the percentage passing into the five statistic areas for the three varieties. The analysis shows that the simulation results using the calibrated parameters, simulation results without considering the rolling friction coefficient, and experimental results have similar trends.

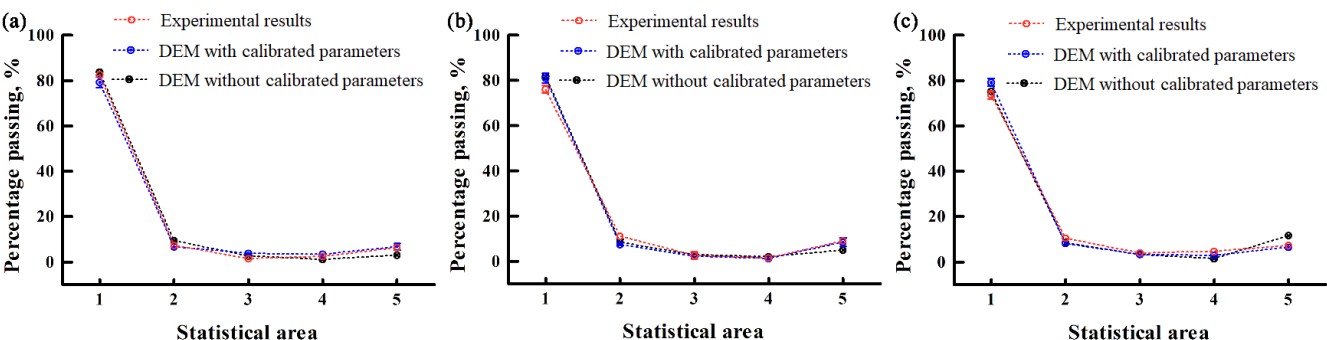

**Figure 21.** Comparisons of the simulation results with the test results of the percentage passing from different statistic areas in the "self-flow screening" for (**a**) SN42, (**b**) JD17, and (**c**) ZD39.

Further analysis of the percentage passing of the simulation and experimental results for the three varieties is shown in Figure 22.

For the SN42, the simulation results using the calibrated parameter was slightly smaller than the experimental results, with a relative error of 0.7%. The difference between the simulation results ignoring the rolling friction coefficient and experimental results was larger, with a relative error of 3.3%.

For the JD17, the simulation results using the calibrated parameter was slightly smaller than the experimental results, with a relative error of 0.1%. The difference between the simulation results ignoring the rolling friction coefficient and experimental results was larger, with a relative error of 4.9%.

For the ZD39, the simulation results using the calibrated parameter was slightly larger than the experimental results, with a relative error of 0.27%. The difference between the simulation results ignoring the rolling friction coefficient and experimental results was larger, with a relative error of 5.23%.

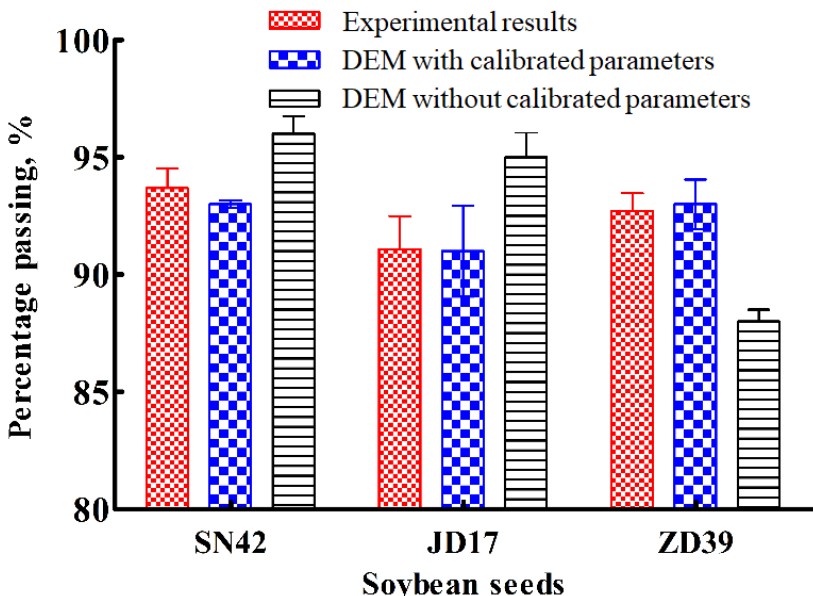

**Figure 22.** The percentage passing of the simulation and experimental results for three varieties.

Therefore, for the self-flow screening test, the simulation results using the calibrated parameters were closer to the experimental results.

Through the comprehensive analysis of the simulated and experimental results of the rotating cylinder test and self-flow screening test, our results showed that, for the three varieties of soybean seed particles, the simulation results using calibrated parameters were closer to experimental values than simulation results without the rolling friction coefficient. At the same time, the simulation results using the calibrated parameters were within the error range of the experimental results. Therefore, the results of the parameter calibrations in this paper were high in accuracy.

## 7. Conclusions

In this paper, the physical parameters of soybean seed particles of different varieties were tested, and the rolling friction coefficients, which could not be measured by test, were determined by calibration methods. The accuracy of the calibration parameters was verified by rotating cylinder test and self-flow screening test. The conclusions are as follows:

(1) The simulation of the repose angle demonstrated that the RFCP-P had a large effect on the angle of repose. The simulation of self-flow screening demonstrated that the RFCP-B had a large effect on the percentage passing. It showed that RFCP-P and RFCP-B needed to be accurately calibrated.

(2) A comprehensive test of the repose angle demonstrated that the RFCP-P had a significant effect on the angle of repose, whereas the RFCP-B did not have a significant effect on the angle of repose.

(3) The RFCP-P was calibrated using a repose angle test. By mixing organic glass spheres with soybean seed particles for the repose angle test, the RFCP-B was calibrated further. The calibrated parameters were verified by means of a rotating cylinder test and self-flow screening test. The results showed that the calibrated parameters were accurate and valid. The RFCP-P and RFCP-B should be considered in the simulation of soybean seed particles.

In summary, parameters are important for simulation. It is particularly important to select and calibrate the relevant parameters accurately. Though the experimental apparatus in this paper is not necessarily applicable to other research subjects, a similar research methodology could be adopted. In addition, there are various methods for modelling particles when the object of study is a different shape. Due to time constraints, only the

multi-sphere method was used to model particles in this paper; therefore, other methods for modelling particles and more in-depth analytical studies on the calibration of parameters should be considered for the next step of research. This paper could provide some reference for relevant areas of inquiry.

**Author Contributions:** Conceptualization, D.Y.; methodology, D.Y.; validation, D.Y. and Y.T.; investigation, K.S. and resources, N.Z. and K.S.; writing—original draft preparation D.Y.; writing—review and editing, Y.W.; supervision, L.Z.; project administration, J.Y.; funding acquisition, D.Y. All authors have read and agreed to the published version of the manuscript.

**Funding:** The authors are grateful to the National Natural Science Foundation of China (No. 52130001) for the financial support of this work.

**Institutional Review Board Statement:** Not applicable.

**Informed Consent Statement:** Not applicable.

**Data Availability Statement:** Not applicable.

**Conflicts of Interest:** The authors declare no conflict of interest.

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
