# Peer review of "Measurement and Calibration of DEM Parameters of Soybean Seed Particles"

_agriculture, doi:10.3390/agriculture12111825_

Round 1
Reviewer 2 Report
The discrete element method has become widely used in many application fields. This, together with the need to generate higher quality data, requires that the calibration of the parameters be increasingly precise. In the manuscript presented for review, the authors studied the rolling friction coefficient between particles (RFCP-P) and particle-boundary (RFCP-B). As these parameters are complicated to measure experimentally, they are usually calibrated. This kind of research, not profoundly original but necessary, allows us to continue advancing in the knowledge of granular materials.
I believe some issues should be addressed but then would have no problems in recommending this article for publication.
Concerns about the presentation:
1- The quality of all figures is poor.
2- There are many typographical and grammatical errors (Among others: repeated labels (line 89 and 92), unit error (l.92); copy-paste errors as the cited [94] in line 124).
3- In section 2.4 there seems to be some missing information: Poisson’s ratio?; Ku is the coefficient? Or cos(\theta): What is \theta?
4- L.147 “Use the PCC software”… something wrong?
5- L.183 What does it mean good appearance?
6- Section 2.6.1. How many experiments did you perform? Did you change the particles used? Did you change the orientation of the particles as you did in the stiffness experiment?
7- Figure 9. What are the dimensions of the particles (individual and as a whole)?
8- L.225. What does it mean “horizontal values”?
9- L. 231. How do you control stability?
10- The scale in figure 12 is not the same in all cases, which makes comparison difficult.
11- L-823. “The RFCP-P is taken as 0.01, 0.02, 0.03, 0.04 and 0.05, and the RFCP-P is taken as 0.02” the second RFCP-P should be RFCP-B?
12- The equations (8) and (9) are already clear in figures 16 and 17, while the results mentioned in lines 310 and 321 are not clear where they come from.
13- Fig. 21 (b) and (c) use different axis orientations, making comparison difficult.
14- L.400. There is a lost figure??
15- Conclusion (2): “ The simulation of self-flow screening demonstrates that the RFCP-B has a large effect on the percentage passing.” is not clear from fig. 22.
16- Reference 23 and 26 is the same.
Reviewer 3 Report
The manuscript entitled "Measurement and Calibration of DEM parameters of soybean seed particles" has been investigated in detail. The topic addressed in the manuscript is potentially interesting, however, there are some main issues which should be addressed by the authors as listed below:
First of all, “Introduction” includes a poor literature review. The applications in the first sentence excludes wide applications (Ref 1-6), for example:
· (2021). Modelling and simulation of fruit drop tests by discrete element method. Biosystems Engineering.
The Introduction section needs a major revision in terms of providing more accurate and informative literature review and the pros and cons of the available approaches and how the proposed method is different comparatively. The novelty of this research with respect to previous research should be clearly stated in the introduction. In an appropriate table, a summarized tabular literature review should also be added. In the manuscript, some relevant papers to particle shape modelling, particle size distribution and contact models could be added as well as more internationally references, for example:
· Structural and micromechanical properties of ternary granular packings: Effect of particle size ratio and number fraction of particle size classes, Materials, 13(2), 339, (2020).
· Shape modelling of fruit by image processing. Commun. Agric. Appl. Biol. Sci. 70(2), 161–164 (2005)
· A novel approach to a realistic discrete element modelling (DEM) in 3D. Commun. Agric. Appl. Biol. Sci. 72(1), 205–208 (2007)
· An Approach to represent realistic particles of bulk assembly in three-dimensional-DEM simulations and applications. Commun. Agric. Appl. Biol. Sci. 76(1), 33–36 (2011)
Second, there are many minor editorial and technical mistakes in the text. Thus, writing of the manuscript could be leveled up by an academic editor. For example;
· Figure and table captions should be clear and precise that are understandable without the text.
· Most of the images are poor in quality.
· And please define all symbols and abbreviations used in the text at first use, list them in a Nomenclature and then use them throughout. Also, all symbols in the text should be italic form.
Third, the methodology is superficial and subpar. For instant, input parameters of the DEM simulation should be summarized in a table. So, the repeatability and reproducibility of the study must clearly be proved with details for other interested researchers. There is no details of software and measurement systems in the manuscript.
Forth, a flowchart of the work is a need.
Fifth, "result and discussion" lacks of broad and in-depth discussion and comparison with other references.
Sixth, conclusions should be improved in standard form including recommendations, future work and pros and cons of the methodology. Additionally, pros and cons of the software could be mentioned.
Seventh, there are several papers from the authors in the reference list on soybeen that should be explained their differences. Also, there are about 20% self-citation in the manuscript.
Reviewer 4 Report
The paper titled: “Measurement and Calibration of DEM parameters of soybean seed particles” is mostly well-written and interesting for readers. The conclusions mostly correspond with the performed research. However, the two main issues should be addressed before this article can be accepted for publication.
1) The novelty of performed research should be emphasized. Authors cite several papers in which soybean seeds were used as test materials for both DEM and Experiment.
2) This paper completely lacks discussion i.e. comparison of the authors finding with the finding of other researchers. Even within cited articles, I see several of them which could be used for that, could probably find even more if searched more internationally.
Below are some minor, mostly language/editing mistakes which should be corrected as well.
Line 53: fric-tion
Line 222: an-gle
Line 261: percent-age
Line 421: soy-bean
Something wrong with symbols 88, 89, 92, 93, 94, 95, and so on.
Line 88: The typical unit used for density is kg/m3 as in Tab. 1
Line 124: ASAE standard [94] is not in the bibliography.
Fig.4: Apart from the fitting curve equation, the coefficient of determination (R2) might be given as well to recognize the quality of fitting.
Lines 145, 146, 147, 184, 200, etc.: Passive voice is usually used in research papers e.g. “Repeat the test 5 times” could be rephrased as: “The test was repeated 5 times.”
Paragraph 2.5.1
Since soybean is non-spherical, the particle can have a non-vertical velocity component/displacement and rotation after the rebound. Some more information should be given about how the authors prepared the experiment and accounted for such energy dissipation in contact.
Fig. 20 x-axis label should be corrected.
Fig. 22 “Simulation results without calibrated” some word is missing or should be rephrased.
Conclusion 1, is a paper summary, not the actual conclusion.
Round 2
Reviewer 1 Report
I am satisfied with the revision of the manuscript. However, I disagree with the statement that the rolling friction coefficient is irrelevant to the shape of particles. See, for example:
Wensrich CM, Katterfeld A. Rolling friction as a technique for modelling particle shape in DEM. Powder Technology. 2012 Feb 1;217:409-17.
Author Response
The author used a repose angle test with a soybean seed particle-glass ball mix to calibrate the RFCP-B, where the description is problematic, and therefore removed "Since the rolling friction coefficient is irrelevant to the shape of the particles and the boundary, " as the reviewer‘s comment,as detailed in line 284-286.
Reviewer 2 Report
The authors did not respond to my comments or did so poorly. Just 2/16 comments could be considered answered.
Concerns about the presentation:
Point 1:The quality of all figures is poor.
Response 1: According to the reviewer’s comment, the author revised the relative figures. Not changes. The figures 2, 3, 8, 11, 12, 13, 19, 21 are still low resolution.
Point 2: There are many typographical and grammatical errors (Among others: repeated labels (line 89 and 92), unit error (l.92); copy-paste errors as the cited [94] in line 124).
Response 2: According to the reviewer’s comment, the author revised the relative part of the paper, as detailed in lines 86-95. Not changes. All the errors still there.
Point 3: In section 2.4 there seems to be some missing information: Poisson’s ratio?; Ku is the coefficient? Or cos(\theta): What is \theta?
Response 3: KU is a coefficient to be determined in the formula for calculating the elastic modulus, the value of which can be obtained by interpolation. Not solved.
Point 4: L.147 “Use the PCC software”… something wrong?
Response 4: According to the reviewer’s comment, the author revised the relative part of the paper, as detailed in lines 137-149. Not changes. Still using PCC without any explanation about what it means.
Point 5: L.183 What does it mean good appearance?
Response 5: “Good appearance” is meant to be intact and unbroken in appearance. Not included in the manuscript
Point 6: Section 2.6.1. How many experiments did you perform? Did you change the particles used? Did you change the orientation of the particles as you did in the stiffness experiment?
Response 6: For the trials, three test specimens of each variety were used to produce three replicate experiments for each specimen. The specimens were made with the soybean in a random orientation with no fixed direction. Not included in the manuscript
Point 7: Figure 9. What are the dimensions of the particles (individual and as a whole)?
Response 7: The authors selected 200 grains of each variety, analysed their dimensions and took the average value to create a 13-sphere model for individual grains. A population of particles was generated according to the law of being normally distributed by volume. Not Solved
Point 8: L.225. What does it mean “horizontal values”?
Response 8: The“horizontal values” meant to levels, it is a scope of the value. Not clear enough.
Point 9: L. 231. How do you control stability?
Response 9: During the simulation, wait for the particles to be generated and then settle for 2 seconds. This method is used to ensure stability. Not Solved. Wait 2 seconds does not ensure stability, you have to measure for example, the kinetic energy.
Point 10: The scale in figure 12 is not the same in all cases, which makes comparison difficult.
Response 10: The result to be compared in the paper is the number of particles. Therefore use different axis orientations is no effect. Not Solved. Not an important issue.
Point 11: L-823. “The RFCP-P is taken as 0.01, 0.02, 0.03, 0.04 and 0.05, and the RFCP-P is taken as 0.02” the second RFCP-P should be RFCP-B?
Response 11: According to the reviewer’s comment, the author revised the relative part of the paper, as detailed in line 271. Not Solved. Nothing changes.
Point 12: The equations (8) and (9) are already clear in figures 16 and 17, while the results mentioned in lines 310 and 321 are not clear where they come from.
Response 12: Each figure is analysed for three varieties. the rest results are come from figures’(b, c). Not Solved. where does the results 23.86º and 18.72 come from??
Point 13: Fig. 21 (b) and (c) use different axis orientations, making comparison difficult.
Response 13: The result to be compared in the paper is the number of particles. Therefore use different axis orientations is no effect. Not Solved. Not an important issue.
Point 14: L.400. There is a lost figure?? SOLVED
Response 14: According to the reviewer’s comment, the author added the figure.
Point 15: Conclusion (2): “ The simulation of self-flow screening demonstrates that the RFCP-B has a large effect on the percentage passing.” is not clear from fig. 22.SOLVED!
Response 15: “ The simulation of self-flow screening demonstrates that the RFCP-B has a large effect on the percentage passing.” is come from fig. 12.
Point 16: Reference 23 and 26 is the same
Response 16: According to the reviewer’s comment, the author delated the reference.Not Solved
Author Response
Please see the attachment.
I am very sorry that the author uploaded the wrong document in the first round of replies and caused you trouble.

Reviewer 4 Report
The authors mostly ignored my previous remarks (including typographical) from the previous round. therefore I can not accept this paper in current form.
1.) This paper still lacks of discussion. Lines 415-433 are summary of the paper, lines 435 - 445 are just the conclusions. None of it is the discussion and comparison with other researchers.
2.) Conclusion 1 is not the conclusion, but summary of what has been done.
3.) Fig. 22 “Simulation results without calibrated” – calibrated of what? Some word describing of what was calibrated is still missing. Or this phase need to be rephrased. Figure should be self-describing without looking into the text.
For more, please see my original remarks.
Author Response

(The authors gave the same response as above.)

Round 3
Reviewer 2 Report
Concerns about the presentation:
Point 1:The quality of all figures is poor.
Response 1: According to the reviewer’s comment, the author revised the relative figures. The figures 2, 3, 8, are still low resolution. That is up to you and the editors.
Point 2: There are many typographical and grammatical errors (Among others: repeated labels (line 89 and 92), unit error (l.92); copy-paste errors as the cited [94] in line 124).
Response 2: According to the reviewer’s comment, the author revised the relative part of the paper, as detailed in lines 86-95. You explain in line 91 \rho_0, m_0 and V_0, it is not necessary to do it again in line95.
Point 3: In section 2.4 there seems to be some missing information: Poisson’s ratio?; Ku is the coefficient? Or cos(\theta): What is \theta?
Response 3: KU is a coefficient to be determined in the formula for calculating the elastic modulus, the value of which can be obtained by interpolation.
Not solved.
Line 125: “elasticity modulus is calculated as follows:”
then, “where E is the modulus of elasticity” is redundant.
Is \mu the Poisson’s ratio? Otherwise, why mention it?
L.132: K_u is it a tuning parameter? Then “K_u is the coefficient” does not seems to be the proper way to mentioned it.
L.133: “calculate cos(\theta)” but it is never mentioned what \theta is?
Point 4: L.147 “Use the PCC software”… something wrong?
Response 4: According to the reviewer’s comment, the author revised the relative part of the paper, as detailed in lines 137-149. Not changes. Still using PCC without any explanation about what it means or citation of the software.
Point 10: The scale in figure 12 is not the same in all cases, which makes comparison difficult.
Response 10: The result to be compared in the paper is the number of particles. Therefore use different axis orientations is no effect. Not Solved. Not an important issue.
Point 13: Fig. 21 (b) and (c) use different axis orientations, making comparison difficult.
Response 13: The result to be compared in the paper is the number of particles. Therefore use different axis orientations is no effect. Not Solved. Not an important issue.
New Point: EDEM software is not cited
Reviewer 4 Report
1) Careful reading for grammar and typos is still required: e.g.: line 340 “snapshaot”.
2) Figs. 2, 3, 8, 11, 12, 13, and 21 are of poor quality.
3) Fig 21. is not corrected as suggested in my previous reviews.
“Simulations results calibrated” -> “DEM with calibrated parameters”,
“Simulations results without calibrated” -> “DEM without calibrated parameters”?
4) Discussion is still weak, but at last, better than in previous versions.
I believe this paper is now acceptable for publishing, what is left is just text and figures editing.
